# Isolation, Characterization, and Unlocking the Potential of Mimir124 Phage for Personalized Treatment of Difficult, Multidrug-Resistant Uropathogenic *E. coli* Strain

**DOI:** 10.3390/ijms252312755

**Published:** 2024-11-27

**Authors:** Alla Golomidova, Yuriy Kupriyanov, Ruslan Gabdrakhmanov, Marina Gurkova, Eugene Kulikov, Ilya Belalov, Viktoria Uskevich, Dmitry Bespiatykh, Maria Letarova, Alexander Efimov, Alexander Kuznetsov, Egor Shitikov, Dmitry Pushkar, Andrey Letarov, Fedor Zurabov

**Affiliations:** 1Winogradsky Institute of Microbiology, RC Biotechnology RAS, Prospekt 60-Letiya Oktyabrya 7 Bld. 2, 117312 Moscow, Russia; alla_golomidova@mail.ru (A.G.); rgabdrahmanov136@gmail.com (R.G.); eumenius@gmail.com (E.K.); ilya.belalov@gmail.com (I.B.); sulfatreduction@gmail.com (M.L.); efsasha@bk.ru (A.E.);; 2Department of Urology, Russian University of Medicine (ROSUNIMED), 2nd Botkinsky Proezd, 5 Bldg 20, 125284 Moscow, Russia; dr.kupriyanov@mail.ru (Y.K.); pushkardm@mail.ru (D.P.); 3Research and Production Center “MicroMir”, Nizhny Kiselny Lane 5/23 Bldg 1, 107031 Moscow, Russia; mgurkova@micromir.bio (M.G.); v.yuskevich@micromir.bio (V.U.); 4Lopukhin Federal Research and Clinical Center of Physical-Chemical Medicine of Federal Medical Biological Agency, Malaya Pirogovskaya ul. 1a, 119435 Moscow, Russia; d.bespiatykh@gmail.com (D.B.); egorshtkv@gmail.com (E.S.)

**Keywords:** UPEC, bacteriophage, personalized phage therapy, Gamaleyavirus, N4-like bacteriophage, O101 O-antigen, therapy-recalcitrant pathogens, antibacterial drug resistance, phage defense systems

## Abstract

*Escherichia coli* and its bacteriophages are among the most studied model microorganisms. Bacteriophages for various *E. coli* strains can typically be easily isolated from environmental sources, and many of these viruses can be harnessed to combat *E. coli* infections in humans and animals. However, some relatively rare *E. coli* strains pose significant challenges in finding suitable phages. The uropathogenic strain *E. coli* UPEC124, isolated from a patient suffering from neurogenic bladder dysfunction, was found to be resistant to all coliphages in our collections, and initial attempts to isolate new phages failed. Using an improved procedure for phage enrichment, we isolated the N4-related phage Mimir124, belonging to the Gamaleyavirus genus, which was able to lyse this “difficult” *E. coli* strain. Although Mimir124 is a narrow-spectrum phage, it was effective in the individualized treatment of the patient, leading to pathogen eradication. The primary receptor of Mimir124 was the O antigen of the O101 type; consequently, Mimir124-resistant clones were rough (having lost the O antigen). These clones, however, gained sensitivity to some phages that recognize outer membrane proteins as receptors. Despite the presence of nine potential antiviral systems in the genome of the UPEC124 strain, the difficulty in finding effective phages was largely due to the efficient, non-specific cell surface protection provided by the O antigen. These results highlight the importance of an individualized approach to phage therapy, where narrow host-range phages—typically avoided in pre-fabricated phage cocktails—may be instrumental. Furthermore, this study illustrates how integrating genomic, structural, and functional insights can guide the development of innovative therapeutic strategies, paving the way for broader applications of phage therapy in combating multidrug-resistant bacterial pathogens.

## 1. Introduction

Urinary tract infections (UTIs) rank among the most prevalent bacterial infections globally, with an estimated 400 million cases and approximately 230,000 deaths reported in 2019 [1]. The substantial burden of these infections is primarily attributed to uropathogenic *Escherichia coli* (UPEC), which accounts for approximately 70–90% of community-acquired UTIs and up to 50% of hospital-acquired cases [2]. UPEC’s pathogenicity is facilitated by its capacity to adhere to uroepithelial cells, evade host immune responses, and form biofilms, resulting in recurrent infections and complications such as sepsis [3]. The escalating challenge of antibiotic resistance among UPEC strains underscores the urgent necessity for alternative therapeutic strategies.

One promising alternative is the utilization of bacteriophages (phages), which are viruses that specifically target and lyse bacterial cells. Phage therapy, which predates the antibiotic era, has regained attention as a potential remedy for multidrug-resistant bacterial infections, including those caused by UPEC [4]. Bacteriophages confer several advantages over traditional antibiotics, including high specificity, the capacity to co-evolve with bacterial resistance, and minimal disruption to the host’s microbiota [5]. Nevertheless, the isolation of phages that effectively target UPEC strains presents considerable challenges.

Bacteriophages have been successfully utilized to combat *E. coli* UTIs, including cases associated with MDR strains of the pathogen [6,7,8]. Although most coliphages exhibit relatively narrow host ranges, infecting only a subset of *E. coli* strains [9,10,11], the isolation of phages targeting a majority of these strains is relatively straightforward from environmental sources such as sewage, open water [11,12,13], and feces from animals [9,14,15], which have proven to be rich reservoirs of novel bacteriophages against various *E. coli* isolates. However, some clinical isolates of UPEC pose challenges in phage isolation, exhibiting an apparently lower abundance of effective bacteriophages in most environmental sources compared to those active against other *E. coli* strains. This phenomenon, while well recognized by practitioners in phage therapy, is underreported in the scientific literature; nonetheless, difficulties in isolating phages for certain *E. coli* strains have been documented [16,17,18].

The host range of bacteriophages, defined as the set of bacterial strains within one or more species that a given phage can infect and form plaques on [19,20], is constrained by various factors. These include variations in cell surface structures and bacterial antiviral systems that inhibit phage replication at different stages of the viral life cycle (reviewed in [21,22]). For *E. coli*, O antigens have been shown to play a significant role in determining phage infectivity, as these polysaccharides can effectively shield the cell surface from phages lacking specific proteins that facilitate penetration through this barrier (recently reviewed in [10]). Concurrently, O polysaccharides (OPS) may serve as primary receptors for many phages [11,23,24]. Many of these viruses (though not all) possess tail spikes with hydrolase or lyase enzymatic activity capable of degrading the OPS backbone (reviewed in [25,26]) or, less frequently, deacetylase activity, which removes lateral O-acetyl groups without degrading the OPS backbone [27,28,29].

The selective pressure exerted by such O-specific phages often results in the emergence of resistant mutants lacking OPS or featuring altered OPS structures, commonly associated with decreased O antigen synthesis [24,28]. These rough mutants frequently exhibit reduced resistance to immune factors [30] (see also review [10] and references therein), but they also become susceptible to certain phages that recognize protein receptors [24,30], provided these phages are not hindered by antiviral systems.

In this study, we present the isolation and characterization of bacteriophage Mimir124, which infects an MDR UPEC124 strain that caused urogenital infection in a patient unresponsive to antibiotic treatment and all commercially available bacteriophages.

Mimir124 phage belongs to *Gamaleyavirus (G7cvirus)* genus of bacteriophages (Viruses; Duplodnaviria; Heunggongvirae; Uroviricota; Caudoviricetes; Schitoviridae; Enquatrovirinae). Members of this genus infect a wide range of Gram-negative host bacteria, using mostly polysaccharide primary receptor molecules on the surface of bacterial cells to initiate infection, such as O-antigens. Some phages of this genus may also use capsular polysaccharides or other polysaccharides or outer membrane proteins as the receptors. O-antigen-mediated adsorption allows *Gamaleyavirus* coliphages to target various *E. coli* strains, including “difficult”, a valuable option for the individualized therapy of patients suffering from infections caused by pathogenic strains of *E. coli* resistant to most of the available ready-to-use coliphage cocktails.

The application of this phage in compassionate therapy resulted in the eradication of the MDR pathogen and subsequent clinical improvement in the patient. Notably, some mutants selected for Mimir124 resistance acquired sensitivity towards siphoviruses, specifically recognizing cell surface proteins LamB and BtuB as terminal receptors, suggesting that the major factor complicating phage treatment of the UPEC124 strain is an effective non-specific surface defense system rather than the presence of some unusual intracellular antiviral systems.

## 2. Results

### 2.1. Case Description

The patient, a 91-year-old male, has been undergoing intermittent catheterization for over a year due to neurogenic bladder dysfunction. During this period, he has been hospitalized four times for severe urosepsis, each time receiving antibiotic therapy with carbapenems. The patient has a history of non-insulin-dependent diabetes mellitus, chronic B-cell lympholeukemia, hypertension, and a history of an acute cerebrovascular event in 2007. Following the most recent hospitalization, the patient was discharged home in early November 2023 and treated with a long-acting beta-lactam antibiotic, which resulted in the elimination of bacteria in urine samples. However, laboratory results of the patient’s biomaterial from 30 January 2024 identified an *E. coli* strain in the urine resistant to a wide range of antibiotics and all commercially available bacteriophage preparations (Table 1). The pathogen isolate has been designated as “UPEC124”.

The analysis of the patient’s urine from 30 January 2024 revealed the presence of leukocytes in a concentration of 82.0 cells/μL, which is above the normal range of 0.0–9.0 cells/μL. In consideration of the patient’s debilitated condition, the attending physicians at the Federal State Budgetary Educational Institution of Higher Professional Education, “A.I. Evdokimov Moscow State Medical University,” a constituent institution of the Ministry of Health of Russia, opted not to prescribe antibiotics. Instead, they requested the Research and Production Center “MicroMir” to select bacteriophages for individual therapy. A sensitivity test of the *E. coli* isolate for all available bacteriophages in the MicroMir Research and Production Center bank was completed on 12 February 2024. The results demonstrated that no bacteriophages were active against the *E. coli* isolate. Consequently, the isolate was transferred to the Winogradsky Institute of Microbiology, Research Center for Biotechnology, Russian Academy of Sciences, where on 10 April 2024, a bacteriophage designated Mimir124 was isolated. Following the completion of the characterization of bacteriophage Mimir124, it was transferred to the MicroMir Research and Production Center for the manufacture of a therapeutic phage preparation under Good Manufacturing Practice (GMP) conditions. On 3 May 2024, the bacteriophage preparation was administered orally three times per day in a volume of 10 mL, with a titer of 10^7^ PFU/mL. On 4 May 2024, intravesical instillations with the bacteriophage preparation were initiated at two-day intervals, with a dosage of 30 mL, with a titer of 10^7^ PFU/mL, administered during catheterization. The results of the examination conducted on 10 May 2024 indicated the total elimination of *E. coli* in the biomaterial obtained from the patient. The urinalysis conducted on the patient on 10 May 2024 indicated the presence of leukocytes in a concentration of 2.3 cells/µL, falling within the normal range of 0.0–9.0 cells/µL.

### 2.2. Bacteriophage Isolation and Primary Characterization

Five environmental water samples were collected from various sources around Moscow and screened for bacteriophages against UPEC124. Phage Mimir124 was isolated from water from the Likhoborka river, Moscow, Russia. This sampling site has previously demonstrated high yields of coliphages, potentially indicating contamination of the water source with fecal matter originating from the abundant wild duck population present at this location. The phage produced clear plaques with a diameter of 3 mm. Occasionally, larger plaques up to 1 cm in diameter with clear centers and very turbid peripheries were observed. These larger plaques appeared after three rounds of phage purification through sequential single-plaque isolation and constituted up to 30% of the total plaques on some plates. Since sequencing (see below) did not reveal any additional phage genomes, we considered the turbid plaques to be a phenotypical variation of the phage. The exact mechanism behind the formation of these two distinct plaque types is unclear and warrants further investigation.

Transmission electron microscopy (TEM) of the sucrose-gradient purified Mimir124 phage revealed that it is a large podovirus with a head diameter of approximately 70 nm (Figure 1). The massive adsorption apparatus was observed around the tail, featuring collar-like elements and tail spikes extending downward. The overall morphology is similar to that of the N4-related coliphage G7C, which we previously isolated and studied [31]. The host recognition apparatus of G7C is composed of 12 branched adhesin molecules ([28] and our unpublished data).

We conducted a search for additional hosts for Mimir124 by testing 7 *E. coli* strains from our laboratory collection, including common laboratory *E. coli* strains missing LPS, environmental isolates obtained from horse feces, and urological clinical isolates. However, no strain was found to be susceptible to Mimir124 other than its original isolation host, UPEC124.

Next, we analyzed the phage’s adsorption kinetics on UPEC124 cells. The adsorption process was relatively rapid and efficient, with almost all phages adsorbing within 3 min, leaving no detectable residual fraction (Figure 2).

The adsorption constant was estimated to be about 2.1 × 10^−9^ mL min^−1^, pointing out the high affinity of Mimir124 bacteriophage towards the surface of UPEC124 *E. coli* cells.

### 2.3. Bacteriophage Mimir124 Genome

Genomic DNA was extracted from the DNAse-treated, sucrose gradient-purified phage sample and submitted for Illumina sequencing. The phage genome was assembled into a single contig of 72,768 bp. The positions of the physical ends of the virion-encapsidated DNA were then determined using the PhageTerm algorithm [32]. The left-end position was found to be similar to that of bacteriophage G7C, where the genomic ends were verified by Sanger sequencing run-off experiments [31], further confirming the accuracy of the PhageTerm output. Mimir124 was determined to have 401 bp-long direct terminal repeats.

An NCBI nBLASTt search using the whole Mimir124 genome as a query identified the closest protein-annotated match as Gamaleyavirus SP5M, with 94.4% of the identity over 85% of the genome length (Figure 2). A comparison with phage G7C, the first described representative of the Gamaleyavirus genus, revealed a similar level of identity (92% nucleotide identity over 83% of the genome length). These results, supported by the virion morphology data, allow us to classify Mimir124 as a species within the Gamaleyavirus genus (Figure 3).

The genomic similarity of Mimir124 to other representatives of the genus was analyzed using the VICTOR algorithm. Phylogenomic GBDP trees [33], inferred from the proportions of identical nucleotides in blast-based alignment as a distance (D6 formula), are shown in Figure 4. The branch lengths of the resulting VICTOR tree are scaled. OPTSIL clustering [34] identified 27 species clusters from 35 input phage genomes, along with a single genus cluster.

The genome was annotated using Pharokka-Phold-Phynteny pipeline [35,36,37,38,39]. The genome encodes 101 ORFs. Notably, it encodes homologs for most of the proteins known to play essential roles in the life cycle of bacteriophage N4 [40], which is paradigmal for the Enquatrovirinae family to which the Gamaleyavirus genus belongs. However, no apparent homolog for gp1, which has been shown to be involved in middle transcription initiation [41], was identified, although the second middle RNA polymerase co-factor gp2 is present.

The genes encoding the host receptor recognition proteins, gp88 and gp89, are located similarly to other Gamaleyaviruses [16,31], between the homolog of N4 phage gp63 (ORF 85 in Mimir124) and the N4 tail protein homolog gp67 (gene 90). The organization of this locus is consistent with that of G7C, aligning with the proposed branched structure of the adhesins based on the TEM data (Figure 1).

A BLASTP search for the closest protein homologs of gp88 and gp89 identified *Klebsiella* RCIP0012 (58% a.a. identity) as the nearest homolog for gp88. For gp89, the best match for the C-terminal 683 residues, which include the putative receptor binding region, was a *Klebsiella* phage protein from the vB_Kpn_ZC2 virus (68% of identity). The N-terminal fragment, responsible for branched fiber virion attachment and putative tailspike (gp88) attachment [28], had its closest homolog in *Escherichia* phage IME11 (92% a.a. identity). Notably, both IME11 [42] and vB_Kpn_ZC2 [43] are also Gamaleyaviruses. These findings suggest that phage Mimir124 encodes a branched adhesin characteristic of most Gamaleyaviruses, consisting of a tail-fiber protein (gp89) that mediates the attachment of the tailspike protein (gp88). Both of these putative receptor recognition proteins (RBP) contain their own receptor binding region (RBR). Given that the best hits for both RBRs were *Klebsiella* phage proteins, we screened the MicroMir *Klebsiella* strain collection for potential hosts of Mimir124. However, none of the xx strains tested positive in spot tests. Thus, *E. coli* UPEC124 remains the only known host for Mimir124 at present.

An HHpred [44] search for gp88 sequences revealed structural similarity to several enzymatically active phage fibers and tailspike proteins. The best match was to the *Klebsiella* phage P560 tailspike, which possesses polysaccharide lyase activity (PDB ID 7VUL_C). Gp89 RBR (a.a. positions 283–926) was also predicted to be structurally similar to enzymatically active phage proteins, with the top hit being the phage Phi92 colanidase tailspike gp150 (PDB ID 6E0V_B). These results suggest that Mimir124′s primary receptor is likely a polysaccharide, such as bacterial O antigen, which may be degraded or modified during the transition from reversible to irreversible adsorption [10].

These findings were facilitated by the significant advancements in bacteriophage genomics and proteomics over the past decade. The application of protein bioinformatics to bacteriophages is now well-supported by an extensive body of structural and biochemical data.

To further explore the interactions of Mimir124 with the host surface, we obtained phage-resistant clones of the host strain. To avoid oversampling the specific mutant type(s) prevalent in the culture, we first plated the UPEC124 strain into individual colonies, then used five individual subclones to select resistant mutants by plating suspensions on phage agar containing a high dosage of the phage (stringent selection conditions). The mutant clones were tested alongside their parental cultures for susceptibility to Mimir124 and to several phages from our collection, previously shown to be blocked by various O antigens. These phages served as probes to test the efficacy of O-antigen-mediated host cell protection [30,45], see also [10]. All five parental clones were resistant to all the phages except Mimir124 (Table 2).

The mutant clones were uniformly insensitive to Mimir124 but acquired the sensitivity to the BtuB recognizing T5-like phage FimX (a mutant of DT571/2 depleted of the lateral tail fibers [46]), as well as to the LamB-recognizing siphovirus 9g [47,48]. The strains also acquired partial sensitivity to the FhuA-dependent phage T5, though the infection was insufficient to produce individual plaques. This acquired sensitivity suggests the removal of non-specific cell surface protection. However, the mutants remained insensitive to the broad-spectrum myoviruses RB49 and Brandy49 [23]. Additionally, we observed that liquid cultures of the Mimir124-resistant clones spontaneously precipitated after standing for several hours without agitation at room temperature or at +4 °C, a phenomenon not observed in the parental strain (Figure 5).

The LPS profiles of the resistant clones were analyzed (Figure 6).

Interestingly, the ladder-like LPS patterns, similar to those observed in the control strain 4s (Figure 6, lane 1), were absent in all cultures. Only a band at the bottom, corresponding to LPS molecules lacking attached O polysaccharide (OPS), and a band with a single repetitive O-unit were detected. However, these bands were clearly visible or even highly intensive only in the mutant clones compared to the wild type in which no bands were visible (though in some gels, a band corresponding to the O-antigen-less LPS molecules was observed at moderate intensity Notably, in some of these clones, the intensity of the second band (LPS with single O-unit, see Figure 6, lane 6) exceeded that of the bottom band. We hypothesized that the structure of the O antigen of *E. coli* UPEC124 might hinder efficient staining of the polysaccharide using the periodate oxidation method, which targets neighboring hydroxyl groups (-OH) on sugar residues. Only the core oligosaccharide (core OS) could be stained; however, in the wild type, most LPS molecules have OPS of variable lengths, thus distributing available core OS between multiple bands, making them invisible. In the mutants where O-unit synthesis or OPS polymerization is blocked, the total LPS is concentrated in one or two bottom bands, significantly increasing their intensity.

To test this hypothesis, we sequenced the genome of the *E. coli* UPEC124 strain and identified the type of O antigen synthesis cluster using serotype finder [49]. The O-serotype was identified as O101. The structure of O101 type OPS according to the ECODAB database (https://nevyn.organ.su.se/ECODAB/list.php; accessed on 25 November 2024) is: →4)bDGalNAc(1->4)aDGalNAc(1→. In this structure (similar to N-acetyl glucosamine polymer chitin), no neighboring C atoms harbor -OH groups available for the periodate-induced oxidation required for silver staining. Summarizing all the data, we concluded that Mimir124 resistance was associated with the loss of the O101 O antigen, consistent with the hypothesis that a polysaccharide serves as the primary receptor, as inferred from the phage genome analysis. Notably, the removal of the O-antigen-mediated cell surface protection rendered the previously resistant UPEC124 strain susceptible to at least two phages non-active against the parental strain. This finding suggests that the rarity of bacteriophages against UPEC124 in the environment is likely due to efficient non-specific surface protection rather than a highly efficient antiviral system encoded by the strain’s genome. We further screened the UPEC124 genomic sequence using the DefenseFinder web tool [50]. The search revealed nine apparently complete putative antiviral systems (Table 3).

## 3. Discussion

Bacteriophages are increasingly regarded as promising antibacterial agents with the potential to mitigate the global crisis of antibiotic resistance among pathogenic bacteria. While numerous reports of positive clinical outcomes from phage therapy (PT) have been published [58,59,60,61], the criteria for selecting PT agents remain poorly defined. Various factors affecting PT efficacy beyond the in vitro activity of a phage, have been recognized [62,63]. However, there is insufficient data to identify specific bacteriophage characteristics that can reliably predict its in vivo efficacy. Therefore, accumulating information about PT outcomes in conjunction with detailed genomic and biological descriptions of the phages used is of critical importance. Cases where single-phage PT is applied or where phages are used sequentially rather than in cocktails are particularly valuable from this perspective.

In this study, we describe the characterization of a Gamaleyavirus phage capable of infecting a uropathogenic *E. coli* strain, which is considered “difficult” to target with phages. Despite the general disadvantages of this genus, representatives as PT agents, such as narrow host ranges streaming from specific recognition of the variable O antigens by enzymatically active phage tail spikes [28] and the high proportion of the resistant clones (see [24,28]), these phages may be effective in individualized PT. Resistant clones selected by surface polysaccharide-specific phages often exhibit increased sensitivity to other bacteriophages (see [10] for review) and may also become more vulnerable to immune factors such as complement [30]. Additionally, the recognition of a primary receptor that is highly represented on the cell surface enables rapid bacteriophage adsorption, which may contribute to a favorable PT outcome.

Our findings contribute to the growing body of evidence that adsorption limitations play a predominant role in determining the sensitivity or resistance of natural *E. coli* isolates to bacteriophages. The activity of the intracellular antiviral systems appears to be of lesser importance for this pathogen compared to other bacterial species [50,64]. Although the antiviral systems encoded by the *E. coli* UPEC124 genome may provide fully functional, robust protection against many viruses, our data suggest that certain phages from the Tequintavirus and Nonugvirus genera can circumvent these defenses, underscoring the critical role of O-antigen in determining phage sensitivity in environmental *E. coli* isolates. Nevertheless, further experimental research is required to quantitatively assess the impact of various factors on coliphage host range determination and to elucidate the mechanisms underlying the phenomenon of “difficult” *E. coli* strains. Our results also suggest that, in naturally occurring *E. coli*, the combination of antiviral systems only partially protects cells and/or offers protection against only certain phages. Given the wide diversity of coliphages [11,65], it should be possible to isolate a phage suitable for therapy against almost any “difficult” *E. coli* strain, highlighting the potential of classical phage therapy based on environmental phage isolates. The fact that “difficult” *E. coli* strains often require narrow-specificity phages ([16] and this work) underscores the importance of making individualized phage therapy available to patients and clinicians as well as the capability to adapt phage formulations to the specific requirements of individual healthcare providers [66].

After this article has been submitted to the journal, a systematic study of genomic factors influencing *E. coli* resistance or sensitivity to a large panel of coliphages confirmed the major significance of the O antigen while antiviral systems were found to have marginal impact [67].

This study has inherent limitations, including its focus on a single patient case involving a “difficult” *E. coli* strain; however, it highlights promising prospects for personalized medicine. The inclusion of the Mimir124 phage in commercial phage cocktails could enhance their therapeutic efficacy by broadening their spectrum of activity.

## 4. Materials and Methods

### 4.1. Phage Isolation

To isolate bacteriophages, enrichment cultures were prepared using water samples collected at five different locations within the city of Moscow, Russia. For each enrichment, 500 mL of unfiltered water was placed in a 1 L Erlenmeyer flask and supplemented by 2.5 g of tryptone (HiMedia, Mumbai, India) and 1.25 g of yeast extract (HiMedia, Mumbai, India). The enrichment cultures were incubated overnight at room temperature with gentle agitation (80 rpm) to promote the growth of bacterial hosts present in the sample and phage amplification on them. No target strain culture was added. Following incubation, 1 mL aliquots were taken, centrifuged in a tabletop microcentrifuge at 13,000× *g* for 5 min, filtered through 0.22 µm nylon syringe filter (Vladipor, Vladimir, Russia), and plated on lawns of *E. coli* UPEC124 strain using the conventional double-layer phage plaque assay technique.

### 4.2. Bacteriophage Adsorption Analysis

The adsorption curve experiment was performed as described in [68]. Briefly, a mid-log phase culture of *E. coli* UPEC124 was inoculated with Mimir124 phage to obtain the final phage concentration of about 2 × 10^5^ PFU mL^−1^. The mixture was incubated in a tabletop dry thermostat shaker (Eppendorf, Enfield, CT, USA) at 37 °C. The 10 μL aliquots were taken at 0, 1, 3, and 5 min post-infection and immediately diluted 100-fold with physiological saline. All samples but time 0 were centrifuged in a tabletop microcentrifuge for 1 min at 13,000× *g,* and 100 μL of the supernatants were plated to enumerate the phages. The sample “0 min” was plated without centrifugation to count the initial phage concentration. The serial dilutions of the reaction mixture were prepared immediately after the experiments, and appropriate dilutions were plated for viable cell counting. The whole experiments were triplicated. The adsorption constant K mL min^−1^ was calculated for the period between 0 min and 1 min using the equation K = ln (P_f_/P_0_)/B_t_, where P_f_—count of free phage, P_0_—initial phage count (as plated from the time 0), B—concentration of bacterial cells CFU mL^−1^, t—time interval, min.

### 4.3. Phage Culturing, Purification, and DNA Extraction

*E. coli* UPEC124 was cultured in LB medium (10 g Tryptone (Amresco, Boise, ID, USA), 5 g yeast extract (Amresco, Boise, ID, USA), 10 g NaCl (HiMedia, Mumbai, India), distilled water up to 1 L). For the plates, this medium was supplemented by 15 g of Bacto-agar (BD, Franklin Lakes, NJ, USA) per 1 L; for the phage double layer technique, a soft agar was prepared, containing 8 g of Bacto-agar per 1 L.

To grow high-titer phage lysate, 50 mL of liquid LB medium in 250 mL Erlenmeyer’s flask was inoculated by 0.5 mL of an overnight culture of *E. coli* UPEC124 strain grown in the same medium. The culture was incubated in an orbital shaker at +37 °C and 200 RPM to mid-logarithmic phase and inoculated with single phage plaque, cut from the double-layer plate. After the evident cell lysis, a drop of chloroform was added to finalize the phage outburst and kill the uninfected bacteria. The lysates were clarified by high-speed centrifugation (15 min at 15,000× *g*, Beckman JA-20 angle rotor) and sterile filtered using 0.22 μm syringe filter.

### 4.4. Selection of the Phage Resistant Clones

An overnight culture of *E. coli* UPEC124 was diluted 100-fold in 3 mL of LB medium and incubated with active aeration for 3 h up to OD at 600 nm of ~0.6. The obtained mid-log culture was diluted and plated with a spreader to obtain isolated colonies growing from single cells. Five colonies were used to inoculate overnight cultures, and the next day, mid-log phase cultures were obtained from each of these clones, as described above. The aliquots of 100 μL of these mid-log cultures were plated undiluted onto the phage agar plates prepared as follows. To prepare phage agar, conventional LB plates were overlaid with soft agar as is normally done for the phage titration, supplemented with 10^8^ PFU of filter-sterilized Mimir124 phage stock per plate (no indicator bacterial strain added).

The plates were incubated overnight at 37 °C, and three individual colonies of phage-resistant mutants were streaked out from each of the parental clones.

### 4.5. LPS Profiling

The LPS profiles were analyzed using SDS polyacrylamide electrophoresis of the Proteinase K-treated bacterial cultures followed by a polysaccharide-sensitive sliver staining procedure. All the protocols used were described previously [24].

### 4.6. Phylogenetic Analysis

The genomes were visualized with clinker software v0.0.27 [69].

The phylogenetic analysis was carried out by the VICTOR web service (https://victor.dsmz.de; accessed on 25 November 2024)), a method for the genome-based phylogeny and classification of prokaryotic viruses [70]. All pairwise comparisons of the nucleotide sequences were conducted using the Genome-BLAST Distance Phylogeny (GBDP) method [33] under settings recommended for prokaryotic viruses [70].

The resulting intergenomic distances were used to infer a balanced minimum evolution tree with branch support via FASTME, including SPR postprocessing [71] for the D6 formula. Trees were rooted at the midpoint [72] and visualized with ggtree [73].

Taxon boundaries at the species, genus, and family level were estimated with the OPTSIL program [34], the recommended clustering thresholds [70], and an F value (fraction of links required for cluster fusion) of 0.5 [74].

### 4.7. In Silico Serotyping

The O- serotype was determined using a complete genome sequence and serotype finder [49].

## 5. Conclusions

The successful isolation and characterization of bacteriophage Mimir124 underscore the critical role of targeted, individualized approaches in addressing multidrug-resistant infections caused by challenging *E. coli* strains. The compassionate use of Mimir124 led to the complete eradication of the multidrug-resistant UPEC124 strain in the treated patient, with subsequent clinical improvement, demonstrating the practical therapeutic potential of narrow-spectrum bacteriophages. The findings highlight the pivotal influence of O-antigen-mediated cell surface defenses in determining phage susceptibility and demonstrate the viability of phage therapy even against pathogens resistant to existing antibiotics and commercial phage preparations. Despite the presence of robust genomic antiviral defenses in the UPEC124 strain, the efficacy of Mimir124 suggests that physical adsorption barriers may play a more significant role in natural resistance mechanisms. These results reinforce the need for expanding bacteriophage collections and developing rapid diagnostic tools for personalized phage therapy. Future studies should aim to elucidate the interplay between phage–host dynamics and genomic defense systems, further advancing the clinical application of phage therapy for refractory bacterial infections.

## Figures and Tables

**Figure 1 ijms-25-12755-f001:**
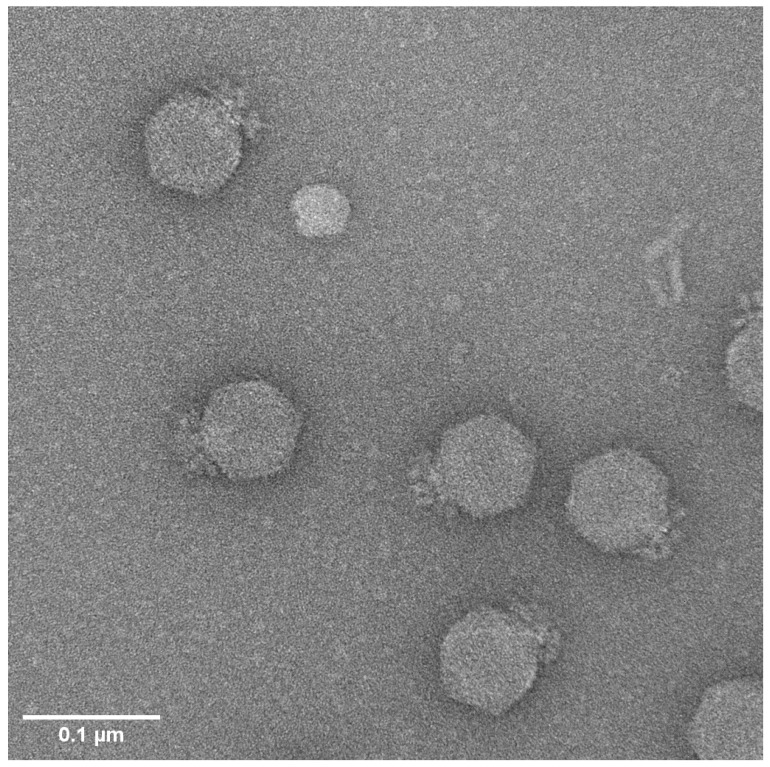
Electron micrographs of bacteriophage Mimir124. Contrasted with 1% solution of uranyl acetate in methanol. Magnification ×90k.

**Figure 2 ijms-25-12755-f002:**
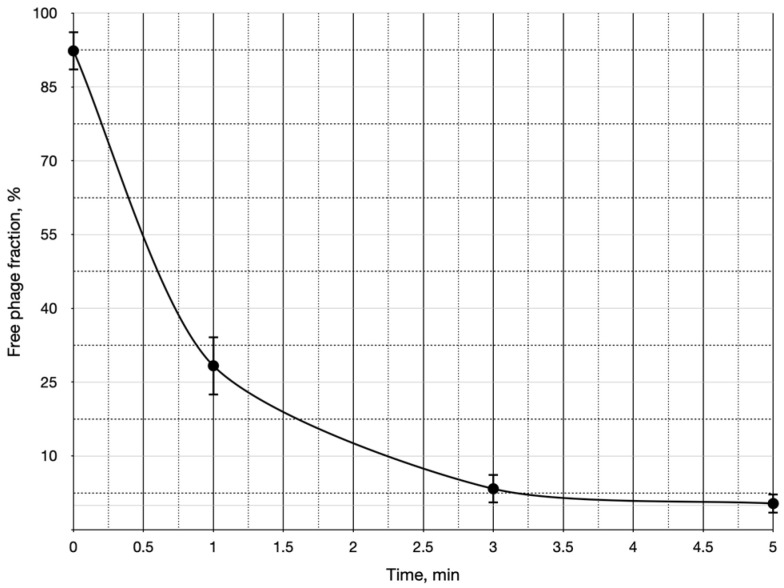
Bacteriophage Mimir124 adsorption curve on *E. coli* UPEC124 cells. The cell density in this experiment was 5.4 × 10^8^ cell ml^−1^. The error bars correspond to adsorption data SD. The experiments were run in triplicate.

**Figure 3 ijms-25-12755-f003:**
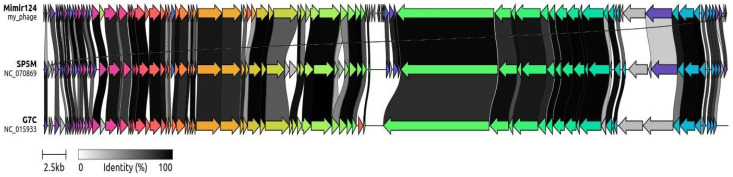
Alignment of the Mimir124 genome to the genome of the closest relative and to bacteriophage G7C (the first characterized representative of the Gamaleyavirus genus). The greyscale code reflects the a.a. identity levels of the encoded proteins. Homologous genes are also shown in same color aspect.

**Figure 4 ijms-25-12755-f004:**
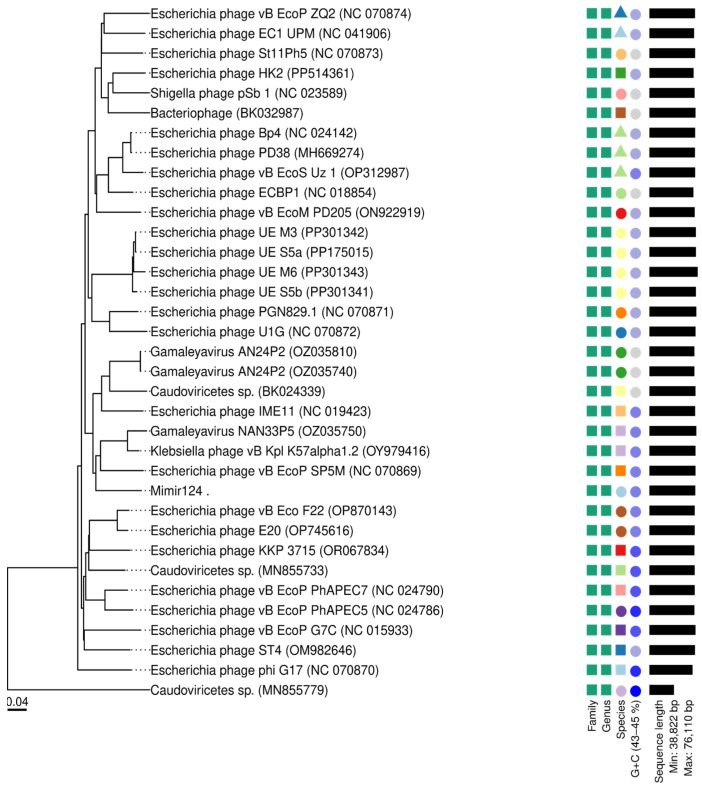
Phylogenomic GBDP trees inferred from the proportions of identical nucleotides in blast-based alignment as a distance function. OPTSIL clusters for taxonomic levels are shown on the right side of the Figure.

**Figure 5 ijms-25-12755-f005:**
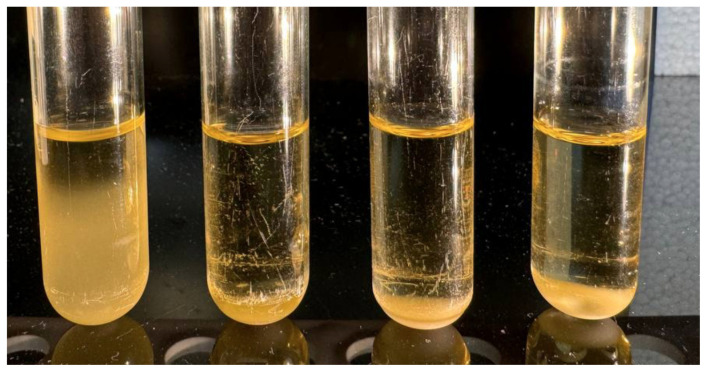
Liquid cultures of UPEC124 clone 1 and its derivative clones selected for resistance to the phage Mimir124 after 10 h incubation without agitation at +4 °C.

**Figure 6 ijms-25-12755-f006:**
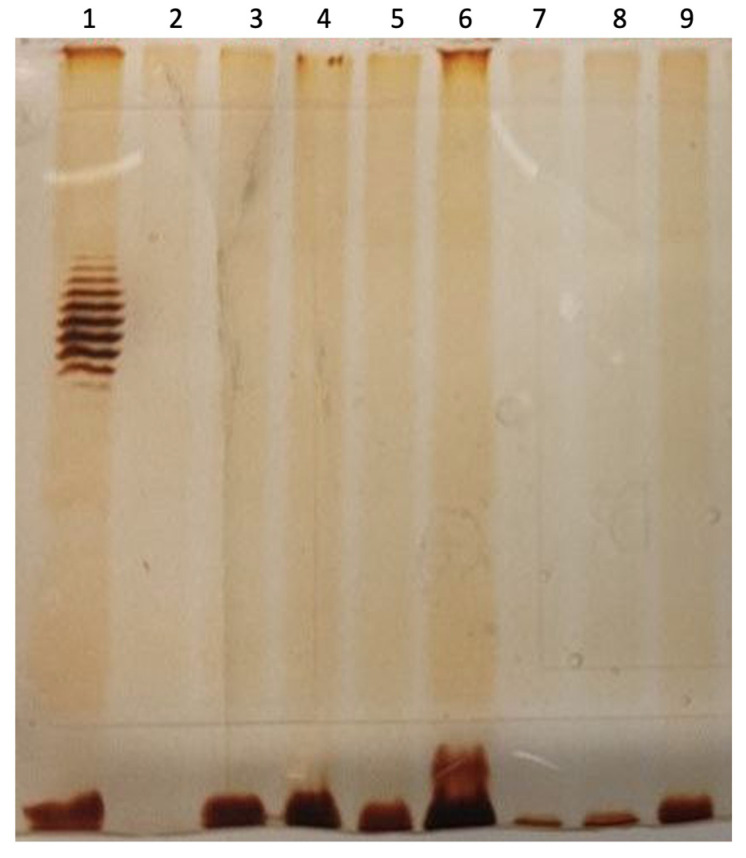
LPS profiling of the *E. coli* UPEC124 and its resistant derivatives. Lane 1 *E. coli* 4s (control of the method), lane 2 UPEC124 wild type, lanes 3–9 Mimir124-resistant clones.

**Table 1 ijms-25-12755-t001:** Pattern of UPEC124 susceptibility to antimicrobials. Susceptibility interpretation: S (sensitive) or R (resistant).

Antimicrobial	Susceptibility
Intesti phage (Microgen, Russia)	R
Coliproteus bacteriophage	R
Complex Pyobacteriophage	R
Polyvalent Pyobacteriophage	R
Ampicillin	R
Amoxicillin/Clavulanate	S
Cefotaxime	R
Nitrofurantoin	R
Phosphomycin	S
Ciprofloxacin	R
Norfloxacin	R
Co-trimoxazole	R
Gentamicin	S
Amikacin	S
Levofloxacin	R
Ceftibuten	R
Ceftazidime	R
Meropenem	S

**Table 2 ijms-25-12755-t002:** Phage sensitivity test of five independent clones of *E. coli* UPEC124 and their derivatives selected for the resistance to the Mimir124 bacteriophage. “+”—the strain is sensitive with the efficiency of plating 0.01–1.0, “+/−”—a growth inhibition zone is seen in the spot of the concentrated phage stock, but no individual plaques are visible in diluted samples. “−”—strain is not sensitive.

*E. coli* Strain	Phages					
	Mimir124	T5	FimX	Brandy49	RB49	9g
UPEC124-1	+	−	−	−	−	−
2.1R	−	+/−	+	−	−	+
2.2R	−	+/−	+	−	−	+
2.3R	−	+/−	+	−	−	+
UPEC124-2	+	−	−	−	−	−
3.1R	−	+/−	+	−	−	+
3.2R	−	+/−	+	−	−	+
3.3R	−	+/−	+	−	−	+
UPEC124-3	+	−	−	−	−	−
4.1R	−	+/−	+	−	−	+
4.2R	−	+/−	+	−	−	+
4.3R	−	+/−	+	−	−	+
UPEC124-4	+	−	−	−	−	−
6.1R	−	+/−	+	−	−	+
6.2R	−	+/−	+	−	−	+
6.3R	−	+/−	+	−	−	+
UPEC124-5	+	−	+	−	−	−
7.1R	−	+/−	+	−	−	+
7.2R	−	+/−	+	−	−	+
7.3R	−	+/−	+	−	−	+

**Table 3 ijms-25-12755-t003:** Antiviral defense systems encoded in the UPEC124 genome.

Defense System Type	Gene Number	Reference
Avs IV	1	[51]
Hachiman	2	[52]
Lamassu-Cap4 nuclease	3	[52]
Gao Hhe	1	[53]
Restriction-modification Type I	3	[54]
Restriction-modification Type I	3	[54]
Restriction-modification Type IV	2	[55]
CRISPR-Cas IE	8	[56]
MazEF	2	[57]

## Data Availability

The original contributions presented in the study are included in the article; further inquiries can be directed to the corresponding author/s.

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
