# Peer review of "Isolation, Characterization, and Unlocking the Potential of Mimir124 Phage for Personalized Treatment of Difficult, Multidrug-Resistant Uropathogenic E. coli Strain"

_ijms, 2024, doi:10.3390/ijms252312755_

Round 1
Reviewer 1 Report
Comments and Suggestions for Authors
This manuscript discuss about Isolation, characterization, and unlocking the potential of Mimir124 phage for personalized treatment of difficult, multidrug-resistant uropathogenic E. coli strain. This manuscript offers valuable insights into using phage therapy for MDR infections, particularly with strains resistant to commercial phages.
Abstract: suggested to add a line about the broader implications of the findings to underscore the impact.
The introduction thoroughly explains the significance of urinary tract infections (UTIs) and challenges with multidrug-resistant (MDR) strains. Adding a brief explanation of the Gamaleyavirus genus and its relevance to E. coli could improve readability for readers unfamiliar with this phage family.
Clear steps are provided for isolating the phage. However, explaining why specific water samples were selected could add context to the methodology
Discussion to further enrich the discussion, consider adding limitations of the study (e.g., generalizability given the single patient case).
Figures and tables enhance clarity, but it might help to provide a legend in Table 2 defining the symbols (+, +/-, -)
Author Response
Comments 1: Abstract: suggested to add a line about the broader implications of the findings to underscore the impact.
Response: Thank you for your suggestion, we have added a sentence to the abstract.
Comments 2: The introduction thoroughly explains the significance of urinary tract infections (UTIs) and challenges with multidrug-resistant (MDR) strains. Adding a brief explanation of the Gamaleyavirus genus and its relevance to E. coli could improve readability for readers unfamiliar with this phage family.
Response: : Thank you for your suggestion, we have added additional information to the introduction section (Lines 92-102).
Comments 3: Clear steps are provided for isolating the phage. However, explaining why specific water samples were selected could add context to the methodology
Response: A context was added in lines 155-157.
Comments 4: Discussion to further enrich the discussion, consider adding limitations of the study (e.g., generalizability given the single patient case).
Response: Thank you for your suggestion, information was added, lines 394-397.
Comments 5: Figures and tables enhance clarity, but it might help to provide a legend in Table 2 defining the symbols (+, +/-, -).
Response: A table legend on lines 285-288 has all the needed symbol definitions.
Reviewer 2 Report
Comments and Suggestions for Authors
This work delves into the isolation of a bacteriophage that can be used in the treatment of bacterial sepsis of Escherichia coli strains with high degrees of resistance. The methodology used and the inference of conclusions from the work is well articulated and well founded.
From my point of view, it should be highlighted that from the point of view of its potential use, it has certain limitations because it is limited to a particular strain of E. coli, and the use of strains with greater scope in the phenomena of bacterial infection would have greater potential.
From my point of view, I consider that a perspective that should be included in the discussion part focuses on work that involves the use of bioinformatics techniques such as docking that enables the characterization of the interaction processes between viral proteins and exposed proteins. on the surface of the cell as the O antigen. The use of genetic editing processes that lead to directed mutagenesis on these phages and molecular dynamics techniques could lead to the design of new phages taking advantage of this type of work.
In my view, adding a brief comment on the potential of protein bioinformatics techniques for future use would be sufficient to allow this work to be published.
Author Response
Comments 1: From my point of view, it should be highlighted that from the point of view of its potential use, it has certain limitations because it is limited to a particular strain of E. coli, and the use of strains with greater scope in the phenomena of bacterial infection would have greater potential.
Response: Thank you for your suggestion, information was added, lines 394-397.
Comments 2: From my point of view, I consider that a perspective that should be included in the discussion part focuses on work that involves the use of bioinformatics techniques such as docking that enables the characterization of the interaction processes between viral proteins and exposed proteins. on the surface of the cell as the O antigen. The use of genetic editing processes that lead to directed mutagenesis on these phages and molecular dynamics techniques could lead to the design of new phages taking advantage of this type of work.
Response: Thank you for your suggestion. While we acknowledge the potential of synthetic biology in advancing phage therapy, we believe this approach is beyond the scope of our current research. The present state of synthetic biology does not yet allow for the rapid and efficient production of novel phages with programmed specificity. Furthermore, conducting docking studies or molecular modeling would require detailed structural data on phage-host interactions, which is currently unavailable for most cases. In the context of patients with acute uroseptic infections, the urgency of treatment necessitates the use of phages isolated and characterized from environmental sources. While we remain optimistic about future advancements that may enable faster and more precise phage development, for now, we rely on classical phage biology methods, which continue to demonstrate robust efficacy in clinical applications.
Comments 3: In my view, adding a brief comment on the potential of protein bioinformatics techniques for future use would be sufficient to allow this work to be published.
Response: A sentence about the impact of protein bioinformatics was added, lines 268-271.